# Olympic Cycle Comparison of the Nutritional and Cardiovascular Health Status of an Elite-Level Female Swimmer: Case Study Report from Slovenia

**DOI:** 10.3390/sports10050063

**Published:** 2022-04-20

**Authors:** Boštjan Jakše, Silvester Lipošek, Nataša Zenić, Dorica Šajber

**Affiliations:** 1Independent Researcher, 1230 Domžale, Slovenia; 2Slovenian Swimming Federation, 1000 Ljubljana, Slovenia; silvester.liposek@um.si; 3Faculty of Kinesiology, University of Split, 21000 Split, Croatia; natasa.zenic@kifst.eu; 4Department of Swimming, Faculty of Sport, University of Ljubljana, 1000 Ljubljana, Slovenia; dorka.sajber@fsp.uni-lj.si

**Keywords:** elite-level athlete, female, swimming, body composition, diet, micronutrients, cardiovascular health

## Abstract

Monitoring the many aspects that are crucial to an athlete’s performance progress is vital for further training planning and for the development of performance and the sport. We evaluated a four-year change (2018 vs. 2022) in the current nutritional and cardiovascular health status of the most successful elite-level female swimmer in Slovenia. Body composition and dietary intake were assessed using dual-energy X-ray absorptiometry and a standardized food questionnaire. The concentration of blood lipids, blood pressure, and serum micronutrients (B_12_, 25(OH)D), potassium, calcium, phosphorus, magnesium, and iron) were measured. The four-year comparison showed an improved body composition status (i.e., increased body mass and decreased body fat (percentage and mass), increased lean soft tissue and total bone mineral density (BMD) (i.e., significantly decreased BMD of a left femoral neck and increased BMD of a spine and head)). We also measured an improvement in the cardiovascular health status of some markers (i.e., decreased total cholesterol, triglycerides, and blood pressure but increased low-density lipoprotein cholesterol), most likely due to the differences in assessed dietary intake (i.e., lower carbohydrate intake, higher total and saturated fat intake, and lower sodium intake). Notably, nutrient intakes that are generally of concern (eicosapentaenoic acid (EPA) and docosahexaenoic acid (DHA), vitamin B_12_ and D, calcium, iron, and zinc (except for fiber intake)) were all within recommended ranges. However, the athlete’s vitamin K and potassium intake were not adequate. Furthermore, in 2018, the athlete did not consume dietary supplements, while she now regularly uses several dietary supplements, including EPA and DHA omega-3, vitamin D, multivitamins, carbohydrate powder, and sports drink. Moreover, from the micronutrient serum, only iron levels deviated from the reference values (37 μmol/L vs. 10.7–28.6 μmol/L). The presented screening example using valid, sensitive, and affordable methods and with rapid organizational implementation may be a viable format for regular monitoring.

## 1. Introduction

A sports training strategy (especially elite-level) represents a long-lasting, systematic, and multifactorial approach and process in which programmed training loads are applied [1]. Monitoring changes in individuals participating in sports and exercise programs (e.g., nutritional, health, performance, and injury/recovery status) enables coaches and athletes to understand better how training, diet, and competition affect athletes when optimizing the training processes [2]. Body composition and appropriate nutrition have essential roles in monitoring the efficiency of physical adaptation to the training process, as well as athletic performance and overall health, aimed at optimizing competitive performance potential [3,4,5].

Various body composition variables contribute to success in numerous sports [6]. However, independently from body composition, elite-level swimmers were found to be taller than those at the sub-elite level [1], which may also apply to female swimmers. For example, in our study published in 2018 [7], the two most successful swimmers within the national team out of 14 at that time were more than 180 cm tall. Furthermore, if we exclude these two swimmers from the analysis, the average body height (BH) of these swimmers’ peers at the time was significantly below 180 cm (i.e., only 171 cm). Moreover, several studies have addressed this question, specifically the relationships between the status of body composition characteristics and performance, and have found that higher-level female swimmers were taller, with lower body fat percentages, with lower body mass indexes (BMI) but with higher muscle mass compared with swimmers competing at lower levels [1,8]. In this regard, for proper interpretation of the obtained results of body composition status, the results of measurements of female swimmers must be analyzed for comparable competitive periods (due to differences in training volume, intensity, consistency in a healthy sports diet, and other aspects of sports life) [9]. In addition, dual-energy X-ray absorptiometry (DXA) in regard to the body composition assessment of athletes in various sports through the competition season is considered to be an accurate, non-invasive, low-cost, and low radiation-emitting method, which can separate the human body into fat, lean, and bone components [10,11].

Optimal nutrition is an essential aspect of an athlete’s training process to improve sports performance and to achieve and maintain optimal health [12]. Furthermore, nutrition is a key determinant of the effectiveness of recovery and adaptive responses to a training process; moreover, it has an important, if not crucial, role in health status [13]. Although optimal nutrition is the key prerequisite for sports performance, the nutrition knowledge of athletes and its impact on their dietary intake, as shown in several review and cross-sectional studies, may be lacking [7,14,15]. Furthermore, in a review article, researchers recognized that micronutrient deficiencies are common in female athletes, particularly for vitamin D, calcium, and iron [12]. However, dietary requirements for swimmers are dependent on swimming style (i.e., freestyle, breaststroke, backstroke, or butterfly), competitive distances (i.e., 50–800 m), training requirements, and competition phase/periodization methods [5,16,17]. Moreover, elite-level swimmers undertake high training volume and frequency (up to three sessions per day), supplemented by dry-land training (e.g., resistance and core training, yoga and flexibility training, running), all of which contribute to the development of the athlete’s performance [17].

Nutritional adequacy may be assessed using dietary intake assessment (i.e., by weighted dietary records, (repeated) 24-h recalls, and using food frequency questionnaires (FFQ)); however, the interpretation of the obtained results should be combined with information regarding serum micronutrients, body composition, and cardiovascular (CV) health status [7] or change. In our previous study on elite-level female swimmers from Slovenia, we recognized the further potential for dietary optimization (i.e., especially regarding energy intake, fiber, polyunsaturated fatty acids (PUFA), calcium, magnesium, potassium, and vitamin D intake inadequacy) that would also benefit their micronutrient (especially for 25-hydroxyvitamina D (25(OH)D) status) and CV health (especially unfavorable blood pressure (BP) profile) status [7]. Furthermore, other studies on female swimmers also reported low energy intake and low intake of several nutrients (e.g., vitamin D, calcium, iron) [17,18].

At the elite level, various major international sports governing bodies (e.g., Fédération Internationale de Football Association and the International Olympic Committee) are performing, requiring, or recommending CV screening [19]. In addition, athletes may have an increased risk of CV disease later in life (e.g., of note, CV diseases may start early in life [20,21,22]), although data on the association between elite-level sports participation and CV health remain largely inconclusive [23,24,25,26]. However, it is unclear whether this tendency reflects specific dietary patterns, cardiometabolic dysfunction, long-term sport activity-associated cardiovascular (mal)adaptations, or both. Nevertheless, for decades, health nutrition has been the cornerstone of CV disease prevention and treatment.

The present case study compares the investigated anthropometric and body composition measures, dietary intake, serum micronutrients, and CV health status of an elite-level female swimmer from Slovenia with her data from the study published in 2018 (four-year period) [7,27]. We chose the case study because only one female swimmer from Slovenia competed in the last Olympics (Tokyo 2020, Japan).

## 2. Materials and Methods

### 2.1. Study Design and Eligibility

The case study protocol was reviewed and approved by the Ethical Board of the Faculty of Kinesiology Split, Croatia (approval document No. 2181-205-02-05-22-003), issued on 8 February 2022, while the results were compared with the data of the study that was reviewed and approved by the Slovenian Medical Ethics Committee (approval document No. 0120-177/2018) and published [7,27]. In addition, the current study was conducted on 14 February 2022 and executed at the same location (Medical Centre Fontana d.o.o., Maribor, Slovenia), using the same protocol and methods as the study completed in April 2018, whose data we compared [7,27].

The athlete was repeatedly recruited through personal contacts with the former national team coach. The swimmer has recently made significant progress in terms of results, so we agreed to repeat the study due to mutual interest. The athlete again signed an informed consent form for inclusion in the study before we applied to the ethics committee for approval. The athlete was not remunerated financially for participation in the study. All the assessments in the study were funded by the authors. The athlete completed the printed questionnaires while waiting for the other measurements (blood sample draw (i.e., 15 mL of blood for complete biochemical assays), anthropometric and body composition measures). Blood assay was collected, and measurements were assessed after an overnight fast.

### 2.2. Subject

The studied elite-level athlete (age 20.5 years) was from the Slovenian female national swim team and was competing in international quality class (World Cups, European Championship (EC), World Championship (WC), Olympic Games (OG)). During both studies (i.e., 2018 and 2022), the studied swimmer was ranked the highest among all national team members.

### 2.3. Outcome

The variables included the detailed characteristics of the athlete’s nutritional and CV health status. The results in the current study were compared with the obtained results from the study completed in 2018 and published [7,27].

#### 2.3.1. Characteristics of the Athlete

The detailed characteristics of the swimmer (i.e., age, education, training status, preferred competitive discipline, qualitative competitive level (International Swimming Federation (FINA) points), ranking in important international competition, type of dietary pattern, menstrual characteristics, swimming motivation, perceived coaching strategy, and training methods) were evaluated with three questionnaires (some variables were part of standardized FFQ [28], others from a modifiable and adaptable questionnaire (which included variables such as swimming motivation, perceived coaching strategy and training methods) [29,30] and some (menstrual status) were collected with a questionnaire developed by the authors). Finally, the qualitative competition level and ranking in important competitions are publicly available information [31].

#### 2.3.2. Anthropometric and Body Composition Parameters

All anthropometric and body composition indices were included (i.e., BH, BM, body mass index (BMI), body fat percentage (BF %), fat mass (FM), lean soft tissue (LST), bone mineral content total (BMC total), bone mineral density total (BMD total) and BMD segmental (i.e., left femoral neck, left femur, legs, pelvis, spine, ribs, arms, and head)).

Body height (cm) was measured using a standardized column scale (Seca 220, Seca Gmbh & Co., Hamburg, Germany), and BM (kg) was measured using a medically approved personal floor scale (Kern, MPS 200K100HM, Kern & Sohn, Balingen, Germany), whereas body composition was assessed using DXA (Hologic, model Discovery W (S/N 87930), with Hologic Apex software, version 13.5.3.1)). Body mass index (kg/m^2^) was calculated from BH and BM. All the assessments were performed by an experienced physician on the same DXA model as in our study conducted in 2018 [27]. 

#### 2.3.3. Dietary Intake

To assess dietary habits in the previous year, we used a manual technique and double-checked it to prevent potential errors: a 52-item qualitative FFQ based on a previously substantiated 50-item FFQ [28]. The FFQ has been tested and validated in a Dutch population for assessing food consumption with seven-day estimated diet records; the results indicated that this FFQ is satisfactorily valid for both genders and different age categories for most food groups [32]. In addition, this FFQ has been used for the Slovene population for athletes (i.e., elite-level female swimmers and artistic gymnasts) [7] and healthy adults [33].

The typical food intakes obtained from the FFQ were calculated by multiplying the frequency of consumption of specific foods by standard portion size (i.e., as suggested by the National Institute of Public Health of Slovenia [34]) and by the nutrients present in one gram. Daily dietary intake was calculated by tallying each food item’s nutrient content. 

To evaluate dietary intake, we used the OPEN Platform for Clinical Nutrition [35], which is a web-based solution developed by the Jožef Stefan Institute in Slovenia [36] and backed by the EuroFIR AISBL [37] and the European Federation of the Association of Dietitians (EFAD). Food intake data (from the FFQ) were used to assess energy and nutrient intake and the frequency of food group consumption. Significantly, it was impossible to estimate sodium, chloride, or iodine intake from food preparation methods (e.g., added iodized salt) based on the FFQ alone, so iodine intake was shown only from the food sources as such. Importantly, FFQ does include minimally processed, processed, or ultra-processed products that include sodium (e.g., mayonnaise, butter, lard, ketchup, confectionery, canned beans, cheese, fries, commercial bread, and pastries) [38]; however, the intake of these foods is usually lower in athletes compared to the general population. The FFQ included dietary supplementation and sports drink (hereafter: supplementation), specifically the name of the manufacturer, the amount of intake, and the frequency of consumption, thus capturing the athlete’s actual dietary intake.

Furthermore, we could also precisely distinguish free sugars from total sugar by using the unique FFQ and OPEN system. All supplementations were included in the evaluation of dietary intake but also discussed separately (i.e., what part it represents in a particular nutrient). Finally, data regarding the dietary intake of the athlete we calculated, were expressed as kcal/d (energy), in units/d (i.e., in g/d (macronutrients), except for dietary cholesterol (mg/d), water intake (L/d), and micronutrients (mg and µg/d)) and percentage of daily energy intake (macronutrients).

In the end, the intake of energy and nutrients of greater importance or concern (i.e., vitamin B_12_, D, eicosapentaenoic acid (EPA) and docosahexaenoic acid (DHA), calcium, and iron) or nutrients that were consumed inadequately/in excess were compared with the reference values for energy and nutrient intake of the National Institute of Public Health of Slovenia [39]; values are summarized according to the recommendations of a Central European (German (D), Austrian (A), and Swiss (CH) (D-A-CH)) reference values [40]. Unfortunately, Slovenian recommendations do not mention the reference values for saturated fatty acid (SFA), monounsaturated fatty acids (MUFA), polyunsaturated fatty acids (PUSA) and cholesterol intake and for EPA and DHA intake; therefore, athlete’s SFA, MUFA, PUFA and cholesterol intake was compared with the D-A-CH reference [40], and EPA and DHA intake with the Dietary Reference Values of the European Food Safety Authority [41].

Finally, water intake from solid foods, beverages, and supplementation (i.e., sports drinks and carbohydrate powder mixed with water) was evaluated. Total water intake was not compared with the guidelines since it depends on the sport, the type of exercise, and the environment [6]. Furthermore, the intake of carbohydrates was compared with the joint position of the Academy of Nutrition and Dietetics, Dietitians of Canada, and the American College of Sports Medicine for nutrition and athletic performance [6]. In terms of energy availability (EA), the variable was not calculated due to the inability to estimate energy expenditure properly or to obtain accurate information.

#### 2.3.4. Serum Micronutrients Concentration

Frequently monitored serum micronutrients that are of concern were assessed and included in the analysis. The following were analyzed: vitamins B_12_ (S-vit B_12_) and D (measured as the vitamin 25(OH)D), calcium (S-Ca), magnesium (S-Mg), phosphorus (S-P) and potassium (S-K), and trace element iron (S-Fe).

For the micronutrients, the Medical Centre Fontana used the same manufacturer and methodology as in our 2018 study [7]. The obtained results from both studies were compared with the following references: for S-vit B_12_ with the reference value suggested to prevent neurocognitive disorders late in life [42]. For 25(OH)D status, three categories were used (i.e., sufficiency: >75 nmol/L, insufficiency: 50–75 nmol/L, and deficiency: <50 nmol/L) [43]. Reference concentrations of serum minerals and trace elements used are from the University Medical Centre Ljubljana, Slovenia, the national laboratory [44].

#### 2.3.5. Cardiovascular Health and Safety Factors

The assessed CV diseases risk factors were total cholesterol (S-cholesterol), high-density lipoprotein (HDL cholesterol), low-density lipoprotein (LDL cholesterol), and triglycerides were measured directly, as was BP. The safety markers included in the blood analysis were uric acid (S-UA), fasting glucose (S-glucose), and hemoglobin. For biochemical analyses, we used the same protocol, manufacturer, and methodology as in our 2018 study [7].

To assess cardiovascular health, the values obtained were compared with the recommended targets for cardiovascular disease prevention by the European Society of Cardiology [45]. For S-cholesterol and HDL cholesterol reference values, the reference values from the national laboratory, the University Medical Centre Ljubljana, Slovenia, were used [44]. S-glucose recommendations were used from the European Diabetes Epidemiology Group for lean adults (BMI < 25 kg/m^2^) [46]. For S-UA, a consensual threshold was used for all healthy subjects [47]. For hemoglobin, we used recommended cut-offs for a non-anemic state from the World Health Organization for non-pregnant females (>120 g/L) [48].

### 2.4. Statistical Analysis

The current study is a case study report of one elite-level athlete and compares the results obtained with athlete from a study four years ago [7,27]; hence, only descriptive statistics are used to present the results.

## 3. Results

### 3.1. Characteristics of the Athlete

The athlete’s characteristics status is presented in Table 1. A comparison of the obtained current results with results from 2018 showed that the swimmer maintained her primary swimming discipline (i.e., freestyle), advanced in FINA points, and improved in competitive results (seven World Cup medals and one European Short Course Championship). In addition, it is essential to emphasize that the athlete maintained the same dietary pattern (i.e., an omnivorous diet). Moreover, the athlete had normal menstrual status and usually trained during menstruation.

### 3.2. Anthropometric and Body Composition Parameters

The athlete’s anthropometric and body composition status are presented in Table 2, while the athlete’s segmental LST are presented in Appendix A. The athlete markedly changed her body composition status in terms of increased BM (+4.1 kg (7%)) and BMI (+1.2 kg/m^2^ (6%)), lower BF % (−2.9% (14%)) and FM (−1 kg (7%)), increased LST (+4.6 kg (10%)), BMD total (+0.08 g/cm^2^ (7%)) and BMC total (+0.33 kg (14%)).

The current results suggest that in terms of segmental BMD, we measured an increase in BMD in specific regions of the body, for example, in the spine region (+43% in BMD spine lumbar and by +15% in BMD spine thoracic region) and head region (+13%); however, a decreased was seen in BMD left femoral neck (−9%).

### 3.3. Dietary Intake

A comparison between the athlete’s daily energy and nutrient intakes is presented in Table 3 and Table 4.

The macronutrient composition of the food intake was 26% fat, 55% carbohydrate, 2% fiber, and 16% protein. The estimated carbohydrate intake of the athlete (5 g/kg BM/d) was below the 6–10 g/kg BM/d or 8–12 g/kg BM/d that is recommended for 1–3 h/d of moderate-to-high intensity exercise (i.e., endurance program) or >4–5 h/d of moderate-to-high intensity exercise (i.e., extreme commitment). We observed this even when considering added energy-type supplementation (i.e., sports drink and carbohydrate powder).

Furthermore, the obtained results showed that the athlete in this study lowered the carbohydrate intake and increased the fat intake (i.e., SFA, MUFA, and PUFA), cholesterol intake, and protein intake. Moreover, in the current study, the SFA (i.e., percentage of energy) and cholesterol intake was higher compared with the results from 2018 and higher than recommended (8% vs. 11% vs. ≤10% of energy for SFA and 189 mg/d vs. 344 mg/d vs. <300 mg/d for cholesterol). In addition, the fiber intake was still not adequate (24 g/d vs. 30 g/d, as recommended).

Significantly, in the current study, the athlete consumed adequate EPA and DHA (785 mg/d vs. 250 mg/d, set as reference). However, the athlete’s EPA and DHA intake with food only was 235 mg/d.

Micronutrients that are often of concern in the general population and among athletes, such as vitamin B_12_ (14 µg/d, the reference is set to 4 µg/d), D (100 µg/d, the reference is set to 20 µg/d), calcium (1095 mg/d, the reference is set to 1000 mg/d), and iron (36 mg/d, the reference is set to 10–15 mg/d) were all adequate.

However, the athlete daily consumed inadequate intake from food only of vitamins C (25 mg/d, the reference is set to 95 mg/d), D (6 µg/d), and E (2.2 mg/d, the reference is set to 12 mg/d), but overall, with the addition of supplements, the intake of those nutrients was adequate.

Furthermore, the athlete’s vitamin K (34 µg/d, the reference is set to 60–65 µg/d) and mineral potassium intake (3690 mg/d, the reference is set to 4000 mg/d) were not adequate.

In line with dietary intake, the analysis showed that the athlete rarely consumed legumes and nuts/seeds (both groups 1–3 times per month), fish, rice, or other grains and cooked potatoes (all three groups once weekly), raw and cooked vegetables (2–4 times per week and less than once a month), more frequently cereals and whole-grain products (both 2–4 times per week).

In 2018, supplements included in the participant’s diet and calculated into the overall dietary intake were occasionally sports drink only. However, in the current study, the athlete included several supplements in her dietary pattern on a regular basis: EPA and DHA omega-3 (with vitamin E included), vitamin D3, multivitamins (with vitamin B_12_, C, folic acid, and iron), and sports drink (2 times daily), with vitamin/mineral-fortified carbohydrate powder (4 times weekly).

Finally, sodium and iodine intake were from food only; therefore, both were underreported.

### 3.4. Serum Micronutrient Status

Serum micronutrients are presented in Table 5. All vitamins and minerals were within reference ranges, except for S-Fe (higher than recommended). Furthermore, S-P was found to be increased compared with the referenced ranges in the 2018 study; however, in the current study, the S-P was found to be within referenced ranges.

### 3.5. Cardiovascular Health

The first observation in comparing the measured results of a blood test is that all the values obtained were within reference values, also S-UA, which was above the recommended range in 2018. A four-year comparison has shown that the athlete lowered S-cholesterol, HDL-cholesterol, triglycerides, S-glucose, and the aforementioned S-UA, while increasing LDL-cholesterol and hemoglobin (Table 6).

## 4. Discussion

### 4.1. Main Findings

To our knowledge, this kind of repeated (i.e., period of the Olympic cycle) screening of the same elite-level female athlete with the same protocol and methods is unique in the scientific literature. With our study, we highlighted five critical findings. First, our results confirmed the importance of screening (i.e., infrequent measurement of selected biomarkers [56]) and analyzing athletes with the same study protocol and methods. The benefits of regular monitoring of elite-level athletes are primarily for the athlete and coach, then for athletes competing at lower-level competitions (i.e., high-performance level and younger athletes) and for the scientific and swimming community. Second, a four-year comparison has shown that the swimmer markedly improved her body composition status (i.e., higher BM, LST, BMC total, and BMD total, lower BF %, FM, and BMD left femoral neck). Third, dietary intake analysis showed that nutrients that are generally of concern among (female) athletes (and female swimmers), such as EPA and DHA, vitamin B_12_ and D, calcium, iron, and zinc (except for fiber), were adequate. However, in addition to the fiber intake inadequacy, the vitamin K and potassium intake were also not adequate. Furthermore, in the study four years ago, the athlete did not consume any supplementation on a daily basis, while the assessment in the current study showed that the athlete consumed several supplements regularly (i.e., EPA and DHA omega-3, vitamin D, multivitamins, carbohydrate powder, and sports drink). Fourth, in regard to the serum micronutrient status, all measured vitamins and minerals were within reference ranges, except for S-Fe (markedly higher than recommended). Finally, markers of CV health status were all within reference values, even S-UA that was higher than the reference ranges. Furthermore, in the current study, compared with the previous one, we have found lower S-cholesterol, HDL-cholesterol, triglycerides, BP, and S-UA, but higher LDL-cholesterol and hemoglobin.

### 4.2. Anthropometric and Body Composition Parameters

A four-year comparison (2018 vs. 2022) in body composition indices measured in the same competitive phase (and same BH) showed a noticeable improvement in BM (62.2 kg vs. 66.3 kg) and BMI (19 kg/m^2^ vs. 20.2 kg/m^2^), BF % (24% vs. 21.1 %), FM (14.9 kg vs. 13.9 kg), LST (45 kg vs. 49.6 kg), BMC total (2.32 kg vs. 2.65 kg), and BMD total (1.09 g/cm^2^ vs. 1.17 g/cm^2^). Importantly, the athlete maintained her BH in both studies (i.e., 181 cm). Regardless, an interesting difference was measured in the BMD of different body segments (especially decreased of femoral neck (0.92 g/cm^2^ vs. 0.84 g/cm^2^ or 9% decrease) and increased BMD of spine (thoracic area: 0.84 g/cm^2^ vs. 0.97 g/cm^2^ or 15% increase, lumbar area: 0.83 g/cm^2^ vs. 1.19 g/cm^2^ or 43% increase) and head (2.08 g/cm^2^ vs. 2.36 g/cm^2^ or 13% increase). In summary, the athlete gained LST, BMD total, and BMC total and decreased FM and BF %, resulting in increased BM and BMI and improved body composition status.

Given that the athlete was four years older in the current study, that she was 16.7 years old at the time of the first study, and that she maintained her BH and competitive discipline during that time, we may safely conclude that the reasons for the obtained differences were primarily due to training development (e.g., coaches probably included various resistance training methods into the training strategy) and secondarily due to improved nutrition in some segments and overall completed physiological maturity.

Our previous cross-sectional study on 14 members of the Slovenian national female swim team showed much lower average BH (173.0 cm), BM (60.4 cm), similar BMI (20.1 kg/m^2^), and higher BF % (22.8 %) [7]. Similar differences in anthropometric and body composition status were seen in a comparison of the athlete’s body composition status results with the results of a US cross-sectional study on 43 female competitive sprint swimmers (i.e., BH: 168.3 cm, BM: 63.8 kg, BMI: 22.5 kg/m^2^, BF %: 25%) [57]. It is noteworthy that the body composition results were presented using bioimpedance and hydrodensitometry in these two studies. Furthermore, the average BMD total (measured by DXA) of the Slovenian national female swim team was 0.85 kg/cm^2^ [30], representing a 38% reduction compared with the current BMD total status of the athlete.

Recently, the researchers conducted a study that included international sprint swimmers (i.e., competed at 50 m or 100 m distances) from eight countries (within the sample 36 females, age 21 years); the paper addressed the relationship between the characteristics of body composition (i.e., measured by bioimpedance) and sports performance. In addition, this study compared the anthropometry and body composition of our athlete and found lower average BH (173.4 cm), similar BM (62.8 kg), but higher BMI (20.9 kg/m^2^), and lower BF % (15.8%) [1]. Notably, the average FINA points of the female swimmers in that international study were 727; their best swimmers scored 910 FINA points [1], while the athlete in the current study scored 921 FINA points. Regardless, the researcher also confirmed a significant and complex relationship between body composition status and sprint performance for both sexes [1]. Finally, a recent study on six elite-level female swimmers (age 21 years) from Japanese members of the national team or qualified in internal qualifications to participate in the preparation of their national training camp showed that their mean BH was 169.2 cm, BM was 60.6 kg, BMI was 21.2 kg/m^2^, and BF % was 17.7% (measured by bioimpedance) [58].

### 4.3. Dietary Intake and Serum Micronutrient Status

The athlete, in terms of dietary intake, relied to a great extent on supplementation (i.e., four years ago, the athlete did not use supplementation regularly) with which she covered the nutritional sufficiency of certain nutrients (i.e., with vitamins/minerals fortified carbohydrate powder, sports drink, EPA and DHA, vitamin C, D, folic acid, and iron).

Furthermore, the estimated average carbohydrate intake of the athlete in previous and our current study (2018 vs. 2022) (5 g/kg BM/d vs. 6 g/kg BM/d) [7] would be barely suitable for low-intensity or skilled-based intensity exercise because 8–12 g/kg BM/d is recommended for >4–5 h/d of moderate-to-high intensity exercise [6]. Carbohydrate intake, especially in relation to a training volume and intensity (i.e., or goals to alter body composition), should be adapted (3–10 g/kg of BM/d) according to the fuel demands of training and the varying importance of undertaking these training sessions with high carbohydrate availability [17]. We assume that the athlete consciously or subconsciously chose overall nutrition with a lower energy intake (2410 kcal/d) and carbohydrate intake (51% of energy) or adjusted energy needs according to the frequency, volume, and intensity of training. In this regard, in the current study, the athlete relied on carbohydrate supplementation based on energy training requirements (i.e., using vitamin/mineral-fortified carbohydrate powder four times per week and sports drink two times per day). Regardless, with this approach, it is likely to be difficult to consume sufficient carbohydrates as recommended for such a projected volume, intensity, and type of workout [6], or the athlete already knows this to such an extent that she effectively modifies her diet to the daily energy training requirements. However, it should be emphasized that our athlete competed in a short-distance discipline (i.e., mostly 200 m); therefore, the overall lower energy intake may still be appropriate or sufficient, at least during the competition phase, when the volume of training is significantly reduced. Although the FFQ includes an estimate of the dietary intake over the past year, the athlete was interviewed after the competitive winter phase. Nevertheless, we acknowledge that in case of insufficient energy availability, the athlete may experience negative health consequences, including hormonal disturbance, menstrual dysfunction, suboptimal bone health, as well as an increased risk of illness and injury [17].

Significantly, while the athlete in the current study increased her protein intake (15% vs. 16% of energy), total fat (21% vs. 25% of energy), SFA intake (8% vs. 11% of energy), and cholesterol intake (189 mg/d vs. 344 mg/d) and decreased the carbohydrate intake (63% vs. 51% of energy) and fiber intake (26 g/d vs. 24 g/d) compared with her results four years ago, the results of the average dietary intakes of the national swam team were very different (protein intake: 13% of energy, total fat intake: 38% of energy, SFA intake: 17% of energy, and fiber intake: 17 g/d, respectively) [7]. Furthermore, consequently, the athlete’s current diet thus contained higher intakes of total and SFA and cholesterol; SFA and cholesterol were higher than recommended [40] and more elevated than assessed four years ago, in 2018.

By severely limiting the intake of raw and cooked vegetables, nuts/seeds, legumes, and grains, in addition to what she otherwise replaces with supplementation, the athlete still has inadequate fiber, vitamin K, and potassium intake. However, our previous study yielded still more reserves in dietary intake as, on average, the national swam team was insufficient in vitamin D, calcium, potassium, and selenium, while intake of SFA exceeded the upper limit [7]. Moreover, based on the completed FFQ and related food groups consumed by the athlete and according to the assessed dietary intake, the conventional dietary intake part contained fewer processed and ultra-processed foods compared to four years ago; therefore, the sodium intake decreased by 23% (2834 mg/d vs. 2307 mg/d), while one part of these foods has been replaced by foods with a higher potassium content (3035 mg/d vs. 3690 mg/d or +22% increase). Hence, the differences assessed in the dietary intake were consistent with obtained differences in the CV health status.

All serum micronutrients, except for S-Fe, were within reference ranges. Notably, the S-Fe value was well above the upper reference range (37 μmol/L vs. 10.7–28.6 μmol/L), possibly due to high iron intake from the source of supplementation (total intake of 36 mg/d, without supplementation only 16 mg/d) or the athletes undergoing additional preventive examination (also to examine the serum ferritin, thyroid function, and chronic inflammation status) to rule out any potential association with preclinical diseases, such as rheumatoid arthritis [59], especially in combination with low triglycerides (more bellow) [60]. Furthermore, in our previous study, S-P was above the reference ranges (1.5 mmol/L vs. 0.84–1.45 mmol/L) [7]; in the current study, it was within the reference ranges (1.1 mmol/L). Of importance, supplementation of vitamin B_12_ and D seemed to successfully translate to adequate serum values (351 pmol/L of vitamin B_12_ and 103 nmol/L of 25(OH)D). 

According to the limited available literature, a study with 85 US female collegiate swimmers (aged from 18–22 years) suggests that the average energy intake was 3229 kcal/d, which is substantially higher (+34%) compared with the current energy intake of out athlete, while the proportions of macronutrients were 31% energy from fat, 54% energy from carbohydrate, and 14% of energy from protein [4]. Of note, the athletes in the compared study were competing at lower levels, and the assessed energy intake ranged from 980 kcal/d to 8050 kcal/d. Furthermore, this study was performed over 15 years ago and, most importantly, the authors did not report the discipline in which the studied female swimmers competed [4]. Regardless, the researchers acknowledge that the energy needs depend on several factors, including type of swim stroke, body mass of the athlete, duration of the activity, and seasonality (training vs. competing) [4]. In the abovementioned recent study on six female members of the Japanese national swim team, the researchers included both sexes (also two males); therefore, these data for the preparation phase (i.e., day 1 to day 3) showed high energy intake (3889 kcal/d to 4447 kcal/d), 26–28% of energy from fat, 53–56% of energy from carbohydrates, and 17–18% of energy from protein [58].

Dietary intake was found to be directly associated with the effectiveness of training, performance, and recovery status among athletes [61,62]. However, the athlete in this study seems to have improved her diet through personal sports development, probably because the various dietary intervention attempts during her career and acquired knowledge of sports nutrition positively impact awareness and knowledge of nutrition [63]. Nevertheless, according to our understanding and the results obtained, there is still a lack of significant improvement in the athlete’s basic diet, which requires more effort, skills, and discipline, including outside the sport itself. In addition, we think this topic (i.e., the athlete’s basic diet) should be more adequately discussed among the experts and less left to the discretion of the swimmer as little progress has been made here.

### 4.4. Cardiovascular Health

All CV health markers were within reference values. Furthermore, the four-year comparison has shown that the athlete lowered S-cholesterol, HDL-cholesterol, triglycerides, and BP, but increased the LDL-cholesterol and hemoglobin as a safety factor. However, some noteworthy differences merit further examination, with several possible explanations for these observed differences. Higher LDL-cholesterol (from 2.2 mmol/L in 2018 to 2.6 mmol/L in 2022) might be explained by increased SFA intake (and refined carbohydrates), which is associated with an increased risk of CV diseases [64,65], whereas dietary fiber (the athlete’s fiber intake was inadequate) yields a reduction in LDL cholesterol (via reduced gastrointestinal absorption) [66].

If we assume that the athlete was healthy at the time of the study, without other accompanying (measured) pathological risk factors, who consumed enough total, PUFA, and EPA and DHA omega-3 within, then currently we cannot suggest a plausible explanation for the obtained low serum triglyceride values (0.4 mmol/L). Nevertheless, according to the underreported analysis of iodine intake due to FFQ limitation (iodine intake was from food only; therefore, there is a possibility that iodine intake was highly underreported regardless of potential significant intake of iodized salt for meal preparation), according to nutrient intake analysis in which the athlete did not regularly consume seafood or the iodine supplementation (which is why we also measured lower sodium intakes and consequently more favorable BP status), we assume that even if the athlete had used iodized salt, she probably had insufficient intake of iodine (the reference is set to 180–200 µg/d). Of note, iodine adequacy only by iodized salt intake may not be achieved without exceeding the permitted reference for daily sodium intake (40% of salt is sodium, while the reference of sodium intake is set to 1500 mg/d) [67]. Regardless, some authors have suggested that the association between low triglycerides values might be associated with certain autoimmunity [60] and/or with elevated S-Fe stores (i.e., S-ferritin) [59]; therefore, for further interpretation, we suggest that the athlete’s S-ferritin status, thyroid function and possibly systemic inflammation status be examined.

Furthermore, the lower BP that was measured in the current study, compared with the results four years ago (115/55 mmHg vs. 128/62 mmHg), may be explained by lower sodium [68,69] and higher plant protein [70], PUFA [71], vitamin C [72], folic acid [73], and potassium intake [69,71]. In addition, it is known that physical activity has beneficial effects on blood pressure [74] and that the athlete was highly active four years ago; however, she currently reported even higher weekly exercise volume (20 h/w vs. 25 h/w in 2018 and 2022). In the 2018 study, the average BP of the female national swim team was 126/73 mmHg [7]. Although physical activity is generally considered an effective treatment strategy for hypertension, some researchers suggest, conversely, that exercise-induced (subclinical) hypertension may also exist, which may equally/substantially increase the risk of cardiovascular events [75]. However, one study analyzed 623 athletes (94 females within the sample) aged 13–77 years and followed them for 10 years; it reported that of the hypertensive women, 92% were aged older than 35 years [26]. Therefore, the likelihood that the frequency, volume, and intensity of training would additionally lower BP is small. This might be true, since the athlete already had a relatively large amount of training four years ago.

Furthermore, our athlete currently experiences an increased LDL-cholesterol to 2.6 mmol/L, which we find disturbing since long-term LDL cholesterol ≥ 2.6 mmol/L is associated with subclinical atherosclerosis, even in the absence of other risk factors [76,77]. Furthermore, this threshold may become a serious health concern or even fatal for athletes later in life.

### 4.5. Strengths, Limitations, and Future Directions

The present case study report examined an elite-level female national team swimmer, currently the most successful in Slovenia. Although the athlete was considered a promising swimmer in the first study four years ago, she was at that time (i.e., 2018) already the best performer in the country. Significantly, in the study, we used the same competitive period, the same location (i.e., medical center), the same protocol, and the same wide data sets of objective methods and compared the athlete’s change/progress in the four-year range. Moreover, the novelty of our results with two screenings of the elite-level swimmer (i.e., with several medals at the most significant competitions) under the same conditions enables valuable interpretations of the results obtained. In addition, this kind of screening or monitoring showed its usefulness as it is affordable, carried out using valid methods, and is not time-consuming.

The study has some obvious limitations inherent to the case study design; therefore, the results should be interpreted with caution regarding transferring the results to other national team members. Furthermore, we are aware when analyzing the dietary assessment of the possibility that the energy and nutrients intake were underreported or under-estimated [78,79], notwithstanding the request for the exact fulfillment of the FFQ by the actual state of nutrition status. The nature of FFQ itself (i.e., completing FFQ by memory), especially for one athlete only, differs significantly from, for example, a three-day (weighted) dietary record. Nevertheless, the athlete did not have access to the completed FFQ from 2018, which showed marked consistency in a similar eating pattern. Moreover, the obtained FFQ results were further considered in the context of body composition and blood tests results. Of note, at the time of the study, our elite-level athlete was not yet using a periodized nutrition plan, as is recommended by most professionals [80].

In addition, there is a lack of follow-up scientific studies on elite-level female swimmers that monitored their changes in various aspects; therefore, there is a need for further studies to investigate the curve of changes in the monitored variables of the athlete that affect the sports performance and development of the sports career.

## 5. Conclusions

A four-year comparison of the results of the elite-level female athlete has shown improvement in body composition indices in all relevant and measured variables. Furthermore, the dietary intake of the athlete also showed several improvements regarding the adequacy of nutrients that are generally of concern among female athletes; however, the improvement was primarily achieved through several supplementations on a daily basis. In fact, the athlete did not regularly consume supplementation four years ago. Regardless, the athlete’s fiber, vitamin K and potassium intake were inadequate. Furthermore, all the measured serum micronutrients were within reference ranges, except for S-Fe, which was markedly higher compared with reference ranges. Finally, the CV health status changes were polarized; S-cholesterol and BP were favorably decreased (also HDL-cholesterol and triglyceride as unfavorably lower), while the LDL-cholesterol markedly increased but was still within the reference ranges.

The screening/monitoring program for a specific set of biomarkers provided several vital insights; however, the successful implementation of monitored methods depends on numerous factors, including the financial cost (need to justify), validity (scientifically established), sensitivity (detecting the difference) [56] and finally, how the head of the athlete’s professional team perceived the proposed monitoring methods. This regards, in our opinion, the proposed simple methods; moreover, the obtained results may provide helpful concrete information for a coach-athlete bond, for swimmers at lower-level competitions (i.e., high-performance level and younger athletes), and for the development of the elite-level swimming in Slovenia.

## Figures and Tables

**Table 1 sports-10-00063-t001:** Characteristics of the athlete.

Parameter	Year 2018	Year 2022
Age (years)	16.7	20.5
Completed education status	Elementary school	Elementary school
Preferred discipline	Freestyle (front crawl)	Freestyle (front crawl)
Beginning training swimming (age)	12	
Swam 8 or more training units weekly (age)	13	
Swam hours per week (h)	20	25
Weekly swam volume (km)	Exceeded 60 km/w	Exceeded 60 km/w
Competition level (FINA points) ^†^		
Long course	832	874
Short course	908	921
High-profile ranking (place)		
Youth EC (400 m, freestyle, 2017)	3rd	
Youth EC (short course, 800 m, freestyle, 2017)	8th	
Youth WC (400 m, freestyle, 2017)	11th	
EC (short course, 200 m, freestyle, 2021)		3rd
EC (short course, 200 m, freestyle, 2021)		6th
WC (short course, 200 m, freestyle, 2021)		4th
OG (short course, 400 m, medley, 2021)		15th
Type of diet	Omnivorous	Omnivorous
Menstrual characteristics ^††^		
First menstruation (age)	15	
Regular (yes/no)	Yes	Yes
Painful (yes/no)	No	No
Perceived intensity of menstruation	Moderate	Moderate
Training during menstruation (yes/no)	Yes	Yes
Oral contraceptive use (yes/no)	No	No
Motivation of swimming ^†††^		
I like swimming	Yes	Yes
I want to be the best and/or win	Yes	Yes
Perceived coaching strategy	Technique was an important part of the training	
Perceived opinion about the content of training	The training was interesting and exciting	
Perceived high-intensity swimming	When repeated sets of maximal intensity	
	When high focus on stroke technique, speed, and force	

^†^ The FINA points table enables comparisons of results among different competitive swimming events. It assigns point values to swimming performances [49]. ^††^ Regular menstrual cycle (eumenorrhea) was defined as 12 cycles per (calendar) year. Moderate bleeding was considered as losing 10 to 35 mL of blood (using 1 to 7 normal-sized tampons or pads) or bleeding from 4 to 6 days [50,51]. ^†††^ Variables with multiple possible answers.

**Table 2 sports-10-00063-t002:** Anthropometrics and body composition status.

Parameter	Year 2018	Year 2022
BH (cm)	181	181
BM (kg)	62.2	**66.3**
BMI (kg/m^2^)	19.0	20.2
BF (%)	24.0	**21.1**
FM (kg)	14.9	13.9
LST (kg)	45.0	**49.6**
BMC total (kg)	2.32	**2.65**
BMD total (g/cm^2)^	1.09	1.17
BMD left femoral neck	0.92	**0.84**
BMD left femur	0.96	0.93
BMD left leg	1.14	1.21
BMD right leg	1.13	1.16
BMD pelvis	1.10	1.06
BMD spine lumbar	0.83	**1.19**
BMD spine thoracic	0.84	**0.97**
BMD ribs left	0.73	**0.80**
BMD ribs right	0.78	0.79
BMD arm left	0.82	0.87
BMD arm right	0.81	0.84
BMD head	2.08	**2.36**

Significant change in value is written bold (i.e., ≥9% of relative change). BH: body height, BM: body mass, BMI: body mass index, BF: body fat, FM: fat mass, LST: lean soft tissue, BMC total: bone mineral content total, BMD: bone mineral density (i.e., all variables in g/cm^2^ units).

**Table 3 sports-10-00063-t003:** Intake of energy and macronutrients.

Macronutrients (Per Day)	Year 2018	Year 2022
Energy intake (kcal)	2433	2262
Carbohydrates (g)	376	**308**
(% E)	62	**54**
Carbohydrates (g/kg BM)	6	5
Total sugars^TS^ (g)	206	**169**
(% E)	34	**30**
Free sugars^FS^ (g)	124	129
(% E)	20	**23**
Starches (g)	153	**94**
(% E)	25	**17**
Dietary fiber (g)	26	24
(% E)	2	2
Fat (g)	57	**66**
(% E)	21	**26**
SFA (g)	22	**29**
(% E)	8	**11**
MUFA (g)	11	**20**
(% E)	4	**8**
PUFA (g)	6	**14**
(% E)	2	**6**
EPA + DHA (mg)	0	**785**
Cholesterol (mg)	189	**344**
Protein (g)	88	97
(% E)	14	**17**
(g/BM)	1.3	**1.6**
Plant protein (g)	29	**40**
(% E)	5	**7**
Animal protein (g)	59	57
(% E)	10	10
Alcohol (mg)	0	0
Total water^TW^ (l)	2.3	**2.7**

Significant changes in value (i.e., 15% of relative change) are written in bold. % E = percentage of total energy intake (general Atwater energy conversion factors were used (kcal/g): carbohydrates and protein = 4, dietary fiber = 2, fat = 9, alcohol = 7) [52]. TS = total sugars: all monosaccharides and disaccharides: free sugars plus sugars naturally present in foods (e.g., lactose in milk, fructose in fruits) [53]. FS = free sugars: all monosaccharides and disaccharides added to foods and beverages by the manufacturer, cook, or consumer (i.e., added sugars) plus sugars naturally present in honey, syrups, fruit juices, fruit juice concentrates, and sports drinks (defined by the World Health Organization [53] and adapted by the Scientific Advisory Committee on Nutrition [54]). SFA = saturated fatty acids; MUFA = monounsaturated fatty acids; PUFA = polyunsaturated fatty acids; EPA = eicosatetraenoic acid; DHA = docosahexaenoic acid. TW = total water: from beverages, solid foods, and supplementation.

**Table 4 sports-10-00063-t004:** Intake of selected vitamins, minerals, and trace elements.

Micronutrients (Per Day)	Year 2018	Year 2022
Vitamins		
Thiamine (mg)	1.9	1.9
Riboflavin (mg)	3.2	**1.9**
Niacin (mg)	48	**22**
Pantothenic acid (mg)	7.6	**6.3**
Vitamin B_6_ (mg)	1.4	**1.9**
Biotin (µg)	54	**35**
Folate/folic acid^FA^ (µg)	353	**1271**
Vitamin B_12_ (µg)	5	**14**
Retinol. equ.^RE^ (mg)	1.1	**0.8**
Vitamin C (mg)	31	**205**
Vitamin D (µg)	10	**100**
Vitamin E (mg)	3.3	**14.2**
Vitamin K (µg)	53	**34**
Minerals		
Calcium (mg)	1447	**1095**
Magnesium (mg)	526	**670**
Phosphorus (mg)	1869	1703
Potassium (mg)	3035	**3690**
Sodium (mg) ^†^	2834	**2307**
Chloride (mg) ^†^	2658	**1095**
Trace elements		
Iron (mg)	12	**36**
Iodine (µg) ^†^	42	**49**
Zinc (mg)	10	**12**
Selenium (µg)	63	62

Significant changes in value (i.e., 15% of relative change) are written in bold. FA = folate/folic acid: folic acid from supplementation to folate conversion factor was used: 0.5 µg of folic acid = 1 µg of folate [55]. RE = retinol equivalents: vitamin A + α-carotene (1 mg retinol equivalent = 12 mg α-carotene) + β-carotene (1 mg retinol equivalent = 6 mg β-carotene) + γ-carotene (1 mg retinol equivalent = 12 mg γ-carotene). ^†^ Sodium, chloride, and iodine intake are from food and supplements only (i.e., without iodized salt). The athlete did not consume many minimally processed, processed, or ultra-processed products or canned products that are included in the FFQ (e.g., mayonnaise, butter, lard, ketchup, confectionery, canned beans, cheese, fries, commercial bread, and pastries) and that include sodium; therefore, the recorded intake of sodium and chloride from food only may be lower than actual intake.

**Table 5 sports-10-00063-t005:** Serum micronutrient status.

Parameter	Reference ^†^	Year 2018	Year 2022
Vitamins			
S-vit B_12_ (pmol/L)	≥258	537	**351**
25(OH)D (nmol/L)	≥75	112	103
Minerals			
S-Ca (mmol/L)	2.10–2.60	2.4	2.3
S-Mg mmol/L)	0.60–1.10	0.9	**0.8**
S-P (mmol/L)	0.84–1.45	1.5	**1.1**
S-K (mmol/L)	3.8–5.5	4.6	4.3
Trace element			
S-Fe (μmol/L)	10.7–28.6	25	**37**

Significant changes (i.e., 10% of relative change) in value are written in bold. ^†^ Serum vitamin B_12_ (S-vit B_12_) reference value suggested to prevent neurocognitive disorders late in life [42]. For 25(OH)D status, we used three categories (i.e., sufficiency: >75 nmol/L, insufficiency: 50–75 nmol/L, and deficiency: <50 nmol/L) [43]. Concentrations of serum minerals and trace elements used are from the national laboratory, the University Medical Centre Ljubljana, Slovenia [44].

**Table 6 sports-10-00063-t006:** Cardiovascular (CV) health and safety factors.

Parameter	Recomm./Refer. ^†^	2018	2022
S-cholesterol (mmol/L)	<5.2	4.4	**3.9**
LDL-cholesterol (mmol/L)	<3.4	2.2	**2.6**
HDL-cholesterol (mmol/L)	>1.3	1.9	**1.6**
Triglycerides (mmol/L)	<1.7	0.6	**0.4**
Blood pressure (mmHg)			
Systolic	120–129	128	**115**
Diastolic	80–84	62	**55**
S-glucose (mmol/L)	<5.8	4.6	**3.9**
S-UA (μmol/L)	<360	377	**340**
Hemoglobin (g/L)	≥120	141	146

Significant changes in value (i.e., 10%) are written in bold. ^†^ Recommendations or reference values: S-cholesterol and HDL cholesterol reference values were from the national laboratory, the University Medical Centre Ljubljana, Slovenia [44]. Low-density lipoprotein cholesterol (LDL-cholesterol), triglycerides, and BP recommendations used were from the European Society of Cardiology [45]. S-glucose recommendations were used from the European Diabetes Epidemiology Group for lean adults (BMI < 25 kg/m^2^) [46]. Serum uric acid (S-UA) consensual threshold is used for all healthy subjects [47]. For hemoglobin, we used recommended cut-offs for a non-anemic state from the World Health Organization for non-pregnant females (>120 g/L) [48].

## Data Availability

The data used to support the findings of this study are included within the article.

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
