# Peer review of "Olympic Cycle Comparison of the Nutritional and Cardiovascular Health Status of an Elite-Level Female Swimmer: Case Study Report from Slovenia"

_sports, 2022, doi:10.3390/sports10050063_

Round 1

Reviewer 1 Report

The novelty of having such a detailed assessment of such a high-level athlete through an Olympic cycle is a unique and interesting aspect of this manuscript. However, I am uncertain that this comparison alone is strong enough to support a stand-alone paper. The FFQ is a valid tool for estimating long-term eating habits, but one would question whether such a high-level athlete was not using a periodized nutrition plan as is recommended by most professionals (Jeukendrup, Asker E. "Periodized nutrition for athletes." Sports medicine 47.1 (2017): 51-63.). This would result in an inability of the FFQ to accurately reflect diet. Further, given the changes seen in diet, it would be important to note if these were gradual or sudden, and in the case of the latter, at what point between 2018 and 2022 they occurred. The timing of nutrient intake with respect to both the competitive calendar and the peri-workout window may also be of interest. Given an n of 1, such details are even more important to consider. Without consideration for such variants, it is difficult to take much away from the report. 

A specific note for table 3: It was confusing to see that the total %E (carbohydrates + Fats + Proteins + Alcohol) did not total 100%. It was 99% for 2018, presumably due to rounding, but 92% for 2022. Additionally, the subcategories do not seem to add up to the totals for %E, though this may be due to forms of the macronutrient not included in the table. Perhaps I am misreading the table, but further clarification may be necessary.

There seemed to be excessive use of the abbreviation i.e. Its use was often contextually inappropriate and it was used parenthetically, which is rarely necessary. For reference, the abbreviation “i.e.” stands for id est, which is Latin for “that is.” This can be taken to mean "in other words" and should only be used when re-stating something, often more explicitly. The abbreviation “e.g.” stands for the Latin phrase exempli gratia, meaning “for example.” 

The content of the paper was made clear to the reader, including the background (introduction), methodology and results + discussion. Still, it would perhaps be beneficial to eliminate some of the redundnacy in reporting the results -between the text and the tables- and not to restate the results within the discussion.

Reviewer 2 Report

Abstract

  • The abstract may benefit from a rearrangement of the results section, possibly showing eventual fluctuations and constant trends of the main variables, possibly coupling it with the competition/non-competition periods. This could be much more informative and helpful to the readers/practitioners.

Introduction

  • Line 45: delete “better”
  • Line 100: “remain”, data is plural.
  • The introduction failed to justify the study design, i.e., why a case study and no other design was used. Please explain it using one-two sentence.

Methods

  • It is not clear why only two assessments per variables were performed. Reading the abstract, I would expect some more within a four-year cycle. Please justify. Both this and the previous comment is crucial to appreciate the novelty of the current study.

Discussion

  • 1: Please merge all sentences into a single paragraph.

The results and discussion are fine if the Authors provide valid justifications for the concerns raised above.
